# Recent Advances of Wearable Antennas in Materials, Fabrication Methods, Designs, and Their Applications: State-of-the-Art

**DOI:** 10.3390/mi11100888

**Published:** 2020-09-24

**Authors:** Shahid M. Ali, Cheab Sovuthy, Muhammad A. Imran, Soeung Socheatra, Qammer H. Abbasi, Zuhairiah Zainal Abidin

**Affiliations:** 1Department of Electrical and Electronic Engineering, Universiti Teknologi, PETRONAS Bander Seri Iskandar, Tronoh 32610, Perak, Malaysia; shahid_17006402@utp.edu.my (S.M.A.); socheatra.s@utp.edu.my (S.S.); 2School of Engineering, University of Glasgow, Glasgow G12 8QQ, UK; muhammad.imran@glasgow.ac.uk (M.A.I.); qammer.abbasi@glasgow.ac.uk (Q.H.A.); 3Research Center for Applied Electromagnetics, Institute of Integrated Engineering, Universiti Tun Hussein Onn Malaysia, Parit Raja 86400, Malaysia; zuhairia@uthm.edu.my

**Keywords:** wearable technology, wireless body area network (WBAN), material, fabrication, wearable antennas, body-centric communication

## Abstract

The demand for wearable technologies has grown tremendously in recent years. Wearable antennas are used for various applications, in many cases within the context of wireless body area networks (WBAN). In WBAN, the presence of the human body poses a significant challenge to the wearable antennas. Specifically, such requirements are required to be considered on a priority basis in the wearable antennas, such as structural deformation, precision, and accuracy in fabrication methods and their size. Various researchers are active in this field and, accordingly, some significant progress has been achieved recently. This article attempts to critically review the wearable antennas especially in light of new materials and fabrication methods, and novel designs, such as miniaturized button antennas and miniaturized single and multi-band antennas, and their unique smart applications in WBAN. Finally, the conclusion has been drawn with respect to some future directions.

## 1. Introduction

The 5G network is on the verge of deployment across many countries, as it demonstrates a great deal of improvement in wireless data rates, connectivity, bandwidth (BW), coverage with the reduction in energy consumption, and latency. The possibility of integration of several wireless devices is expected in future [1,2]. The Internet of Things (IoT) is a fast-emerging technology, which intends to transform and seamlessly connect the world by heterogeneous smart devices. The IoT influences every aspect of life and business, especially healthcare applications [3]. Various kinds of wearable devices are predicted and assumed to be an important part of IoT [4]. According to Ericsson, in [5], nearly 28 billion smart devices will be used by 2021, especially in the field of security, automation, and different electronic systems [6]. Moreover, according to the CISCO report of the Virtual Networking Index (VNI, 2014–2019) [7], the data stream from wearable devices will rise tremendously to 277 petabytes per month in 2019. Wearable devices will increase to 578 million by 2019. Humans or animals are expected to wear these wearable devices and they will have enough capacity to communicate with each other directly. Further, they communicate with other remote devices through wireless modules, which contain a sensor, microcontroller, and an antenna system.

As aforementioned, wearable technology has countless applications; however, WBAN has received important attention due to the widespread applications in health care, sports, battlefield, emergency operations, and care of the elderly and underprivileged children [8,9,10,11,12,13]. Currently, the wireless communications technology demands miniaturized planar antennas with wide BW to support wireless devices, such as Industrial Scientific Medical (ISM) and Wireless Local Area Network (WLAN), etc. [14]. These advanced wireless technologies need high performance and miniaturization that can be achieved by compact antenna systems [15]. Moreover, due to limited continuous radiation exposure on the body, miniaturization of an antenna is a suitable option in the WBAN system because it occupies less space on the human body, either on or off the human body. However, it is difficult to miniaturize a wearable antenna while maintaining its high performance. In order to maintain a simple antenna design with wideband (or multiband) capabilities, easy integration and flexibility have become real challenges and require serious consideration in advanced wireless communications systems [16]. Thus, designing wearable antennas for the wearer’s outfits, several aspects are specifically required to be considered. First, the coupling between the antenna and the human body needs to be investigated to stop the degradation in the performance. The next issue is to cope with the wearable antenna deformation on the body, as this significantly degrades the performance. In addition, to obtain the robust performance of the wearable antennas in various conditions, such as humidity, temperature, and distance between the body and antenna, is also challenging. Numerous parameters, like high electrical and mechanical properties, low cost, lightweight, low loss, flexibility, convenience for wearers, and high resolution/accuracy in fabrications methods are also very important.

Researchers have focused on conventional designs using textile materials and inkjet/screen printing methods [17]. However, they have not been able to solve all the existing challenges previously mentioned, despite showing some acceptable outcomes [18,19,20,21,22,23,24,25,26,27,28,29,30,31,32,33,34,35,36,37,38,39,40,41,42,43,44,45,46,47,48,49,50,51,52,53,54,55]. Thus, it is imperative, in retrospect of the recent progress in wearable antennas and their applications in the WBAN systems. In earlier review papers, various studies focused on antennas that were designed to be flexible. They used a textile substrate and a large planar structure to hide the antenna within cloth or garments (some of them focused on the antenna geometry and its realization in different applications [56,57]). Other studies focused on the mechanical stability and characteristics of the materials, including their impact on antenna design performance [58,59]. A few other researchers examined the influence of the human body on antenna performance and the specific absorption rate (SAR) using different approaches [60,61]. Despite all these efforts, the fabrication methods for the design of wearable antennas showed complexity, less precision, and a tedious process [62]. Unlike conventional antennas, the wearable antennas should be examined in a complex real-life environment, such as the existence of the human body, bending, wrinkling of outfits, as well as washing, etc. However, some of them have been tested in a complex and real-life environment using different shapes and techniques, such as health, sport, and battlefield, etc. Table 1 summarizes various applications of the wearable antennas.

Many review papers like [64,65,66,67,68,69,70,71,72,73,74,75,76,77,78,79,80,81,82,83,84,85,86,87,88,89,90,91,92,93,94,95,96,97,98,99,100,101,102,103,104,105,106,107,108,109,110,111,112,113,114,115] discussed wearable antennas thoroughly. There are many new high-performance conductive and substrate materials, such as MXene ink, small negative meta-composites, polymer/zirconia nanocomposite, and polymer gel, that have been used in various fabrication methods of antenna design. However, previous reviews did not cover these latest materials. Various new fabrication methods are gaining popularity, like tape nanolithography, direct cutting methods, such as cutting plotting and infrared laser for induced graphene (LIG), and direct handwriting methods, such as pen, pencil, or brush paint, etc. However, there are many new developments in the area of antenna miniaturization, like array button antennas, reconfigurable button antennas, miniaturized feeding topologies, circularly-polarized button antennas, wideband button antennas, modular button antennas, dual-band button antennas, and a mosaic antenna that has not been reviewed before. Finally, new unique smart applications are discussed, such as worn safety shoes, a self-tuning antenna for rescue operations, and unique logos emphasize countless applications of the wearable antenna in the WBAN systems. Furthermore, human body parts have been found good to participate experimentally in the working of such antennas. Such a unique experimental application has never been reviewed before. This review article covers all the recent trends in the wearable antennas regarding the materials, fabrication methods, designs, and advanced applications in WBAN. This review paper is organized as follows: after an introduction, in Section 2, various types of materials used for the wearable antennas are discussed. Section 3 describes the various types of fabrication methods. Section 4 describes various types of wearable antennas, such as miniaturized button-shaped and single- and multi-band antennas. In Section 5, recent unique smart applications of wearable antennas with examples are described in detail. Finally, Section 6 provides the conclusion and future direction.

## 2. Flexible Materials

A wearable antenna uses two types of materials, conductive and substrate. These materials should be considered carefully to design wearable antennas using their electrical and mechanical properties. Conductive materials can be divided into two groups: rigid and flexible conductive materials. The rigid conductors can be further divided into various groups, such as copper, silver, and aluminum [116], whereas the flexible conductive materials are smart textile, ink, liquid, graphene (FG), graphite (FGF), carbon nanotube (CNT), and polymers [117]. In addition [118,119], polymer composites are widely used for the wearable applications due to their low loss, flexibility, and stretchability like PDMS-coated silica nanoparticles [120,121], and polymer yarns [122], etc. Substrate materials can be divided as a printed circuit board (PCB) and flexible substrate. PCBs can be further classified into various groups, such as Roger, FR4, and Teflon, etc., whereas flexible substrates are textile, paper, polymers, rubber, foam, etc. [116]. In wearable antennas, a precise characterization of a material is necessary to determine an effective design. There are a large plethora of materials; one of them is the characterization in the WBAN systems [123,124]. In order to define the electrical and mechanical properties of flexible materials, various methods can be used [125,126]. In the era of innovative and disposable electronic systems, the flexible and stretchable materials are excessively in need due to their foldable, bendable, and stretchable properties [127].

### 2.1. Conductive Materials

The wearable antenna uses a ground and top radiator. Conductive materials can be evaluated in terms of their resistivity, conductivity, deformability, weather-proof, tensile strength, and their integration with flexible materials. Various rigid conductive materials have been used for the wearable antennas. These materials have high conductivity, low cost, and can be integrated with textile substrates using adhesive laminated materials without using embroidery and sewing methods; however, their rigid structure restricts their use for flexibility [128,129]. Various smart textiles (e-textile) can be used for wearable antennas, which provide high flexibility and can be sewed into clothing using fabric yarns, such as Shieldit, Zelt, Flectron, taffeta [130,131], Kevlar, nylon, nickel-plated, and silver-plated materials [132], as shown in Figure 1a. These e-textiles are anisotropic and possess low conductivity (1 × 10^6^ S/m), and can be designed using various methods in [133,134]. However, the stretching and crumpling of the wearable devices results in minor cracks, which reduces the performance of the wearable antennas. Various expensive and complex composites make wearable antennas incompatible with respect to low cost given the complexity in the fabrication methods, as shown in Figure 1b. Additionally, the unique features of conductive polymers, such as electrical, mechanical, biodegradability, and recyclability and hydrophobic properties, have enabled them to be used in various applications such as sensors, light-emitting diodes, fuel cells, and batteries, etc. [135]. Moreover, polymers are low lossy materials with high stability in wet conditions due to their low moisture absorption rate, which makes them a suitable choice for wearable applications [136]. Various high-performance materials, such as FG [137] and FGF [138], can be easily incorporated into polymers to improve the tensile strength [139]. In addition, these high conductivity materials can be embedded in polymers without using any complicated technique.

Researchers in [140], embedded conductive fibers in a polymer substrate to design flexible, robust, and frequency-tunable wearable antennas. However, these types of wearable antennas are easily prone to stretch due to the deformation issues in flexible materials. Recently, high stretchable materials have received high consideration to overcome these critical challenges [141,142]. Additionally, the stretchable antennas are suffered due to their low radiation characteristics when they stretched. Various stretchable doped materials can be used, such as liquid metals [143], CNT [144], silver-based fluorine rubber [141], stretchable fabric [145], polymer-zirconia nanocomposite [146], and polymeric nanocomposites (MXene Ink) [147], etc. Various new materials can be used to increase the performance of the stretchable wearable antennas such as doped oxides with indium, fluorine, tin, meshed wired, and transparent fabrics, etc. [148,149]. Recently, a new Ti3C2Tx/PVDF metacomposite with a negative permittivity (−550 to −400) and polymer-zirconia nanocomposite has aroused great attention due to their unique properties. It has opened new directions in the field of wearable applications [150,151]. In addition, new composite materials, such as a graphene oxide layer combined with poly 3,4-ethylenedioxythiophene: polystyrene sulfonate (PEDOT: PSS), can be utilized [152]. Therefore, the mixing of high-conductivity composites with low filler contents can increase the electrical and mechanical properties of the materials, like polymer, as shown in Figure 2. Table 2 summarizes the properties of various flexible conductive materials.

### 2.2. Substrate Materials

Substrate materials mainly support the conductive part of an antenna to improve the radiation characteristics. Specifically, they are also important in wearability. However, most flexible materials possess their low permittivity with low loss tangent, which further increases the performance of the wearable antennas. Numerous flexible substrates have been investigated for wearable antennas such as silk, wool, denim, Cordura, fleece, felt, and so on [156,157]. In [158], textile substrates were used for the antenna designs like a button and a zipper shaped. Additionally, paper materials were also used for the wearable antennas using the inkjet and screen-printing methods. As presented in [159], the paper material was used for a wearable antenna due to its low cost and easily accessible. Various polymer materials such as polyimide [160], PET, PDMS [161,162], and liquid polymer (LP) [163,164] were used for flexible and transparent antennas because they provided high flexibility, low loss, and thicknesses. Moreover, polymer materials show good electrical and mechanical properties, which give them the ability of deformation in wearable antennas [165]. Among flexible substrates, polymer substrates are commonly used for the wearable antennas or in complex designs, like liquid metal antennas [166,167]. In [168], Kapton showed high flexibility as well as soldering tolerance in the antenna design and can withstand the highest temperature like inkjet-printed antennas. The polymer substrate can easily hold the deposited conductive materials [160,163], and conductive inks [162,169]. In [170], a flexible coplanar-fed multi-band antenna was designed on a Kapton polyimide substrate at 2.45 GHz and 5.8 GHz. In [171], a highly flexible and stretchable PDMS layer was fabricated using the plasma bonding technique, as shown in Figure 3a. Gels are quite soft and flexible 3D polymer networks with small-sized pores, which can be used in various applications. In [172], a new type of polymer gel was presented, which present a low level of pores size. Additionally, it can be used in various applications to provide surface uniformity in the materials or bonding purpose. In [173], stretchable polymer materials were obtained by mixing them with different stretchable materials, shown in Figure 3b. In [174], wearable antennas were deposited on various substrates like polycarbonate (PC), polyethylene naphthalate (PEN), and polyvinyl chloride (PVC). Thus, the exact characterization of substrate materials is necessary before designing an antenna. Table 3 summarized the properties of flexible substrate materials. In the design of the wearable antennas, flexible materials have received significant attention as compared to rigid materials.

In summary, the flexible substrates and conductive materials have received great attention instead of rigid materials because the wearable antennas are operating near the body. They are required to be robust against bending, stretching, and twisting, along with low loss. Moreover, they should be comfortable for the users and can be easily incorporated into clothing. However, in [175], polymer materials have received considerable attention due to their unique electrical and mechanical properties and are easily incorporated into various high-performance conducting and substrate materials, such as zirconia ceramic, liquid crystal, or liquid metals, and different composite materials, like MXene ink, etc. Table 4 shows the properties of various flexible materials.

## 3. Fabrication Methods

Fabrication methods play a key role in the design of wearable antennas as they determine the accuracy, cost, and time-consuming process. Recently, new fabrication methods have been used for the design of wearable antennas, such as 3D inkjet, infrared laser cutting, and direct handwriting methods. A brief review of these fabrications methods is given below.

### 3.1. Screen Printing and Gravure

Screen printing is a simple method, which has been used by the electronics industry [177]. In this method, a squeegee blade is driven downwards that forces the screen to contact the affixed substrate. It ejects the ink forcefully via screen-exposed areas on the substrate so that the required pattern can be designed [178]. Moreover, polyester, and stainless-steel materials are also used in this method. Recently, three types of screen-printing techniques are being used, including rotary, flatbed, and cylinder methods. Among them, the flatbed is a standard method. However, screen printing is a simple and additive method as compared to chemical etching due to their low cost and simple fabrication. Various transparent and RFID flexible antennas have been designed using this technique [179,180,181]. In [156], a compact E-shaped antenna was fabricated on polyester fabric using a screen-printing method, as illustrated in Figure 4a.

The polyester fabric was found to be a water-resistant material, and it provided an efficient design for wearable applications. It provided a measured gain of up to 3.5 dBi for WiMAX. In [182], a screen-printing process was used to design a patch antenna, and the sheet resistance was obtained up to 0.5 Ω/sq. In [183], a highly conductive patch antenna was designed using composite materials like graphene, polyaniline (PANI), and PDMS on textile material. Furthermore, an active and screen-printed antenna was designed on non-woven material (Evolon) [184]. The conductive ink was equally spread on the substrate material. The porous polyurethane web was also used to design a robust antenna that could be washable and minimize the moisture or swelling effects. Numerous RFIDs and transparent wearable antennas were fabricated using screen printing [105]. Despite being a simple technique, this method faces certain issues, like limited thickness and low resolution of the printed pattern. Moreover, layer uniformity is another major issue, as thermal curing of ink solvents leaves traces, and it changes with ink viscosity and surface energy of the substrate. Similarly, the gravure printing method is another promising printing technique, as shown in Figure 4b. This technique can be used to transfer ink by just touching the design on the surface of the material. This method gives high-resolution patterns, adjustable depth, and printing area, with low cost. Moreover, this method can be used to print on different materials like rigid, flexible, absorbent as well as non-absorbent materials. Presently, transparent electrodes and their interconnects have been gravure printed using nanowires and CNT on flexible substrates, which are briefly described in [185].

### 3.2. Flexography

In this method, an image is generated using the printing process, as shown in Figure 5a. The protuberance surface of the printing plate matrix is filled with ink, whereas the recessed area is free of ink. For the image printing, the protruding matrix surface contains ink and it touches with substrate material [186]. This method has shown high throughput, clear resolution, and low cost. This method has played a key role in the design of RFID antennas. Moreover, it requires a low viscosity conductive ink, and dry films (thickness < 2.5 µm). Additionally, substrate parameters, such as surface hydrophobicity, porosity, and energy of the material surface, have a direct impact on the printed ink film thickness [187]. Thus, uniformity of the ink thickness and line width can be affected due to the sheet resistance. Table 5 shows the different types of screen-printing methods.

### 3.3. Inkjet Printing or Direct Ink Writing

In this method, the antenna and the RF circuits depend upon the highly conductive inks. The inkjet printers are operated by dropping a small size of ink droplets, such as a few picolitres to produce clear and accurate shapes. This method uses conductive inks like silver nanoparticles. The inkjet printing method can be divided into two main types such as continuous inkjet and drop on demand, as shown in Figure 5b [188,189].

In drop-on-demand, the heads of the inkjet apply pressure pulses to ink with any piezo or thermoelement to drive a drop from a nozzle whenever it is needed. However, the quality of the printing depends upon the ink-like particle size, viscosity, and surface tension. Moreover, this method controls the volume of ink droplets from the nozzle. Additionally, it is the fastest, most economical, and cleanest solution for designing [190,191]. Although the power of a printed antenna is 40% lower than other traditional antennas, it fulfils the ISO 14443 standard at 13.56 MHz [192]. Additionally, the performance can be improved using the extra input power. In [193], silver ink was used to design a dual antenna on a flexible Kapton substrate (∈_r_: 3.5, tan δ: 0.002) using a DMP 2831 printer. In [194], a patch antenna was designed using cotton fabric material at 2.4 GHz. A polyurethane paste was used to avoid direct printing on the cotton material due to the surface irregularity and it minimized the risk of substrate damage. Thus, to avoid the post-heating process in the inkjet method, another sintering and deposition method was used in [195]. These methods use a water-soluble solution for silver ink, which can be easily printed on flexible materials. Apart from that, textile substrates can be used for printing the e-textile conductor using denim, fleece, felt, and polymer substrates. These materials provide low surface resistance (0.49 Ω/sq) after bending. In [196], a wearable tracking system was designed on polyester and cotton substrates. Additionally, a Dimatix printer was used to print the electronic system on these substrates. Moreover, the chemically-cured conductive ink was utilized, which changed the printed method dramatically by reducing the cost [197]. This new method can be used as an inkjet printer to print a silver nanoparticle (AgNP) conductive layer without using any post-processing. Additionally, when the printed antenna was designed using chemical curing of AgNP, the substrate surface then required further treatment so that the nanoparticles could create the conductive trace at room temperature [198].

3D printing is an additive direct writing method like inkjet, and an antenna’s complex structure, such as microfluidic channels, can be designed on the substrate utilizing the FDM (fused deposition modelling), as shown in Figure 5c. In [199], Ninja Flex was heated to print the antenna. In [200], galinstan (GaIn) and Ninja Flex materials were utilized to design an IF antenna. The galinstan liquid was injected in the substrate channels to create a flexible radiator. In [201], the EGaIn was 3D printed utilizing VeroClear material to design the antenna with slots. In the meantime, in [202], a Visijet M3-type Crystal was 3D printed using the EGaIn radiator to design a planar helical antenna. Thus, the substrate was 3D printed, and the LM was created into its cavities using the vacuum-based filling method. However, this technique can be encouraged up to 8 GHz, since the higher-band antennas needed high precision. In [203,204], inductive coils of RFID tags made of GaIn were printed on paper, plastic, latex, and textile materials. With a 200 µm conductive thickness and accuracy of up to micron, this method showed a possible solution for the future liquid antenna. In [205,206], a 3D printing of LM with a thickness of ~2 µm has shown. The 3D printer has shown an ultra-thin conductive LM with high reliability and repeatability and shows the state-of-the-art among LM antennas. Table 6 shows the different kinds of inkjet printing methods.

### 3.4. Stitching, Sewing, and Embroidery

The stitching and sewing process is an important method for a textile antenna. It can be used to design various metallic components for the antenna, such as vias, quarter-wave transformers, or transmission lines or interconnections using conductive fabrics like threads and yarns [209]. In [210], the stitching and sewing technique was used for connecting copper and conductive polyester-taffeta fabric on the substrate to design a planar inverted-F antenna (PIFA). In [211], conductive threads were used to form conductive paths on fabric, like cross-stitch embroidery, as shown in Figure 6a. In [212], the effect of the sewing method was studied for the stability of resonance via a few stitching phases. It was concluded that loose sewing affects the overall performance due to the variation. Embroidery is another traditional method, which is used to design wearable antennas using colored textile threads. In this method, the antenna design can be embroidered on textile materials using the conducting threads. With the developments in technology, computer-aided embroidery machines are now available to design embroidery antennas, as shown in Figure 6b [213]. However, the conducting threads must be strong enough and flexible enough to avoid any damage during the stretching and bending process. Apart from the advantages, the embroidery method has some challenges, which include appropriate conductive threads, strength, flexibility, and continuous shape. In [176], a conductive yarn fabric was used to weave the antenna patch on fabric materials using digital embroidery machines. Additionally, in [214], this method was exploited to design a patch antenna for E-tag applications.

In [215], a spiral-shaped antenna was proposed for conformal applications. In this antenna, the Elektrisola-type e-threads were embroidered on a Kevlar material using a sewing machine. In [216], the FSS type structure was used on the ground to improve the performance of the embroidery antenna. Recently, a dipole antenna is designed using the following satin and contour-filled patterns for WBAN [213]. This satin-based design provided high dimension accuracy and maintained the stability of the resonance frequency. The same design worked for contour fill pattern; thus, it reduces the losses due to the proper alignment between the threads and the current flowing. In [217], a dual-band PFIA antenna was designed for short-range communication at 900 MHz and 2.4 GHz. In [218], a multiband antenna was designed for body-centric communication at 10 MHz and 2.45 GHz, respectively. This design showed an L-shape, and a slit was utilized for the off-body at 2.45 GHz. Moreover, it connected with an electrode and mounted on-body to provide wireless communication at 10 MHz. In [145], a new technique was adopted using silver nanoparticles in active textile materials at low temperatures, which yielded a surface resistivity of 0.2 Ω/Sq. In [104], an E-shaped antenna was designed at 5 GHz with an embroidered E-type microstrip patch on the top side and also an embroidered ground on the bottom side. This antenna used cross-stitching five-filament copper threads on a polyester fabric, and it could be sewed on a felt substrate. The flexibility of the conducting yarns and the precision of the embroidery machines have proved the importance of the embroidery-based antennas [219].

### 3.5. Adhesives

Adhesives are simple fabrication methods, as illustrated in Figure 6c. They can provide the interconnections between the antenna components like vias, connectors, substrates, cables, and conducting parts. They are helpful for the manufacturing and interconnection of fabric antennas to fix the flexible conductive parts on the substrates. Heat-activated adhesives are being used on the backside of the conductive fabrics, like Shieldit. Moreover, adhesives are usually activated through ironing. Additionally, their geometrical dimensions are maintained in the attachment process using water-soluble foil so that they can be used for temporary stiffening and mechanical stabilization [211]. Recently, a tape nanolithography method was used to provide a high-resolution pattern on the substrate and conductive films to realize the nanophotonic structures on adhesive tapes. Moreover, this method provided high flexibility in controlling the number of thin layers and material compositions [220]. Table 7 shows a comparison of fabrics and adhesive methods.

### 3.6. Direct Cutting

Cutting-plotting is a low cost and simple method, which can be used to design wearable antennas. Initially, it was used for the design of UHF RFID tags, made of adhesive copper tape. In [222], a cutting-plotter method was used for the first time by the researchers, which used non-woven conductive fabrics (NWCFs). In [223], SRR type antenna was designed for wearable applications using the cutting-plotting method. However, the precise cutting of geometries with extremely small dimensions was found difficult. Similarly, the automatic robotic with liquid dispenser method was used for the design of wearable antennas [182]. However, laser cutting is an alternative advanced fabrication method for the design of wearable antennas. In [224], the laser-cutting method was used to design a textile UWB antenna via Epilog Zing 24 or Trotec Speedy 300 laser systems. In [225], a picosecond-pulsed laser was used to achieve precise dimensions of the e-textile sheets. In [226], a copper foil was used on textile material before using a laser machine to ensure accurate cutting via the LPKF Proto-Laser S series. Figure 7a shows direct cutting methods of a graphene paper [174].

Recently, an infrared laser method was used to convert carbon precursors directly into graphene, called laser-induced graphene (LIG) [227], as shown in Figure 7b. The LIG can be formed by laser irradiating several carbon precursors including polyether-sulfone, polyimide, wood paper and clothing, etc. The LIG provides a high-resolution and direct-writing method, which shows the utilization of LIG in sensor and wearable technology. Table 8 shows the direct cutting methods.

### 3.7. Direct Handwriting (DHW)

Pen and spray types are new and simple handwriting methods, which can be used for the design of wearable antennas. In the pen method, a conductive ink, like titanium carbide (Ti_3_C_2_), can be used, which is also compatible with different techniques such as painting, stamping, and printing. MXene ink is used in commercially available pens for direct writing the pattern either manually or automatically via AxiDraw. Moreover, pens can quickly write on various substrates, such as paper, textiles, and polymers. The MXene-deposited ink can act as a passive circuit, which is made of silver, copper, and nanoparticle inks [230,231,232,233]. Numerous new methods are encouraged from time-to-time using handwriting, like brush pen [234,235], pencil [236,237,238,239], fountain pen [240,241], and ball pen [242,243,244], have been currently developed to write the wearable electronics directly, and which can be used for various wearable applications. Thus, DHW with a pen is an easy and rapid way of adding conductive materials on substrates to design wearable electronics for the WBAN systems, electrochemical sensing [245,246,247], electronic modules [248,249], and energy storage appliances [238,239], as shown in Figure 8a.

Recently, a spraying printing method was used for liquid metals to design the wearable antennas, as shown in Figure 8b. Apart from that, it is a low cost and a simple fabrication method. In this method, the liquid metal was combined with smart textiles to obtain a pattern [250,251]. In [252], various conductive shapes were designed using the spray method, but these shapes were not utilized within the RF scope. In [173], a low-cost stencil mask for antenna design was shown using high-end methods for drop-casting of silver nanowires. However, the researchers have proposed a spray method for the antenna designs in the patient [253]. Various types of spray antennas have been suggested in patents to design the 3D-printed forms [254,255]. In [250], a theoretical and experimental approach for the spray method was presented to create the wearable devices. Recently, in [256], “the research’s in Drexel university” successfully designed a MXene antenna utilizing the spray method, which worked like a copper antenna. Similarly, a transition metal titanium carbide (MXene) prepared by a single-step spray coating for the antenna design. The design provided a ~100 nm thick translucent MXene antenna with a reflection coefficient (S_11_) of less than −10 dB. By increasing the antenna thickness to 8 mm and S_11_ reached up to of −65 dB [257]. Moreover, a 2D metallic niobium diselenide (NbSe_2_) was used for the antenna design that provides a sub-micrometres thickness of 855 nm, which is less than the skin depth of any other materials. Therefore, the 2D transition metal dichalcogenides may be a good candidate for the rational design of flexible, miniaturized, frequency-tunable, and omnidirectional RF antennas in WBAN systems. In addition, the NbSe_2_-based antenna can be designed using the economical spray method with less time consumption in processing. Furthermore, the transition metals NbSe_2_ antenna provides high, flexibility, low thickness, and dimensions. Additionally, it assists in space communication systems due to the high temperature (~3K) available in deep interstellar space [258]. LM patterning can also be achieved by pressure spraying atomized microdroplets of conductive fluids on a substrate, as reported in [259]. Therefore, the fact that spray coating is a useful technique for printing small to medium size antennas using any substrate material. Table 9 shows a comparison of various DHW methods.

In summary, it can be concluded that the fabrication method is a challenging task because they require high accuracy, low cost, and ease of integration in WBAN systems. In this section, various types of fabrication methods are presented to design high-performance, complex, as well as low-cost wearable antennas, such as 3D printing and laser direct cutting methods. Recently, direct handwriting methods have received considerable attention due to their low cost, mass production, and ease of fabrication. However, these methods are not yet fully matured in the RF field due to the high conductive inks or liquids required to fulfill the demand for future applications in WBAN systems, such as 5G and 6G technologies.

## 4. Wearable Antenna Designs

Wearable antennas are used for wearing either on humans or animals. They operate near the human body, which affects various parameters, like impedance mismatching and detuning effects. Currently, various studies have been conducted to design various types of wearable antennas. Examples of antennas include the monopole planar antenna [261], planar inverted-F antenna (PIFA) [262,263], microstrip patch antenna [264], magneto dipole antenna [265], substrate-integrated waveguide antenna (SIW) [266], electromagnetic bandgap (EBG)-type antenna [267], dipole antenna [268], fractal patch antenna [269], and cavity slot monopole antenna [270]. Similarly, a Yagi-Uda antenna was also used for WBAN in the millimeter-wave (mm-wave) band [226]. A wearable antenna is always encouraged to yield a small form factor in WBAN systems [271]. Additionally, they should be multiband, flexible, and stretchable, so that they can be used for various applications. The antenna miniaturization is usually related to physical and electrical properties. These miniaturization techniques can be divided into two groups, such as material and topology method. Each group contains various types and shapes, such as space-filling curves, meander lines, fractal shapes, metamaterial surfaces (MT), and high dielectric materials [102]. Recently, button-shaped antennas and single and multiband antennas using MT are widely used in the WBAN systems. These techniques have minimized the coupling effect between the body and antenna and reduced the backward radiation. Thus, miniaturization techniques resolved all the existing challenges and maintained the compactness of the design. This section briefly explains button antennas and single and multiband wearable antennas.

### 4.1. Button Antennas

The button antenna is a new design, which can be mounted on textile materials at various locations on-body. The button antenna is usually a rigid surface, which promises a better radiation characteristic as compared to textile antennas [272]. In [273], a cuff button antenna was designed for WLAN, which included a conductive G-shaped patch. It was connected with a cylindrical base by a cylindrical metallic tube and mounted on a Velcro substrate. This antenna provided an omnidirectional pattern. In [274], a button antenna was designed on a pair of jeans for rescue operations at 2.4 GHz and the HiperLAN bands. This design was used to study body propagation, such as the line-of-sight (LOS) and non-line-of-sight (NLOS) channels. In [275], a new wideband button antenna was proposed at 2.4 GHz using characteristic mode theory, as shown in Figure 9a. The design demonstrated excellent performance in free space with BW, gain, and radiation efficiency of 658 MHz, 1.8 dBi, and 97%, respectively, whereas for on-body, it achieved the BW, gain, and radiation efficiency of 788 MHz, 5.1 dBi, and 71%, respectively. Additionally, SAR of 0.45 W/kg at 20 dBm (input power) was obtained. In [276], a novel circularly polarized button shape antenna was designed for a broadside pattern at 5.47–5.725 GHz (UNII Band), as shown in Figure 9b. The design was based on a button-shaped FR4 substrate with a radiating element on both sides. On the top side of this substrate, a large and a small radiating circle was used with the cylinder being fixed on the small circle, while, on the bottom side, a rectangular radiating strip was used. Moreover, at the other side of the rectangular strip, a feeding probe was connected to the coaxial feed underneath the conductive fabric ground. A BW of 610 MHz was obtained in the range of 5.47 to 6.08 GHz, while an axial ratio (AR) BW of 3 dB was obtained within the frequency range of 5.4 to 5.81 GHz. Moreover, the radiation efficiency and the gain of 79% and 3.5 dBi were obtained, respectively. In [277], a cuff-shaped button antenna was designed at the ISM and UNII bands, which was a circular patch with a diameter of 22.3 mm. The shape of a cuff button was conceived from a square-shaped polytetrafluoroethylene (PTFE) taconic ceramic material and its dielectric constant was considered 10. The ground was used on the bottom side to avoid the mutual coupling between the body and antenna. The design provided a high efficiency, high gain, and a VSWR of 94%, 1.5, and 2.1 at both operating band of 2.4 GHz and 5.5 GHz, respectively.

In [278], a new idea was proposed for the design of modular geometry, which acted as a snap-on button and detached the radiation parts easily. All radiators could be used in the same feeding technique. First, a detachable patch achieved a right-hand circular polarization (RHCP), and then left-hand circular polarization (LHCP) by changing the radiator at 5 GHz. Secondly, the design showed a PIFA-antenna by interchanging the resonator at 2.4 and 5.3-GHz. A 3 dB AR BW was obtained from 4.95 to 5.08 GHz for the LHCP and 4.95 to 5.06 GHz for the RHCP, respectively, thus both were centered at 5.01 GHz. Finally, all results showed that the proposed module was less affected by the human body. In [279], a miniaturized spiral Inverted-F antenna was proposed with the omnidirectional radiation pattern. This structure consists of various components, such as spiral line, stub, metal flange, metal ring, and a ground coplanar waveguide (GCPW)-type feeding. The two vias were used at the ends of the spiral line and the feeding stub. The metal ring was printed at the bottom side of the substrate, as shown in Figure 9c. Additionally, the gain on-body at the lower and upper band was −0.6 dBi and 4.3 dBi, respectively. However, the limited SAR of 1.6 W/kg was obtained at 26.4 dBm (input power). In [272], a novel dual-band button shape was proposed for the WLAN band. It showed a quasi-monopole pattern for on-body at 2.4 GHz, while a quasi-broadside pattern band for off-body at 5 GHz. The design showed an efficiency of 91.9% and 87.3% with a measured gain of 0.27 and 4.73 on both bands, respectively. The SAR was checked on various parts of the human body, which was below the limited value (0.20–0.22 W/KG). Currently, a new miniaturized array button antenna was designed, as shown in Figure 9d. It covered 4.5–4.6 GHz for the IoT applications, whereas the 5.1–5.5 GHz for the WLAN applications. Moreover, the design showed the lowest SAR in all these bands along with the S-parameters and total efficiencies were almost similar to the free space communication [280]. Thus, it shows the robustness of the proposed button antenna, as shown in Figure 9d.

In this section, various types of wearable button antennas have been studied, such as semi-flexible and fully-flexible, with unique shape and size of the button antennas. Further, button shapes can be mounted on any parts of the human body. Additionally, these designs provide miniaturization, stable performance, less coupling effects, flexibility, and a high degree of physical robustness using both rigid and flexible materials.

### 4.2. Miniaturized Single and Multiband Wearable Antennas

A wearable device is an important component in tracking and monitoring systems, especially ones using miniaturized multi-band antennas with circular polarization, as they show stable radiation characteristics during bending, stretching, and crumpling conditions [281]. Researchers are working to determine new ways of miniaturization, which could have a low profile and provide stable radiation characteristics under bending conditions. These wearable antennas are also using various communication systems, such as the Global Navigation Satellite System (GNSS). They require circular polarization (CP) and broadband patterns, like GNSS with GPS. Similarly, Galileo, GLONASS, and Compass also provide similar global coverage [282]. These systems are used for various applications, such as rescue operations, military, and health systems. Moreover, they can be used to integrate with cellphones, cameras, wristwatches, computers, and wearable outfits, etc. [283]. Usually, satellite systems need CP and multi-bands at the receiving side to obtain constant polarization and to avoid the time-varying orientation between transmitting and receiving signals for off-body communication. In [284], the antenna was designed for outdoor applications at the GNSS L1/E1 band, which provided CP. Recently, in [285], a CP sleeve badge antenna was designed for the 2.4 GHz ISM band, as shown in Figure 10a. The design was small enough so that it could fit within the jacket sleeve badge. It provided a wide beam-width with a 5.6% BW, RHCP, and a gain of 4.71 dBi. The SAR was limited to 0.34 W/KG. In [286], a dual-band wearable textile antenna was proposed for the ISM (2.22–2.48 GHz) and HiperLAN (4.95–5.80 GHz) applications. The antenna was based on a suspended plate. The rectangular radiator was used as cylindrical vias, slits, and slots to obtain the dual-band resonance and broad BWs. The design provided a unidirectional radiation pattern because of the ground and provided the isolation between the body and antenna. The total efficiency was obtained between 67% and 89%, whereas the gain and SAR obtained were 8.33 dBi and 0.14 W/kg on the chest, and a further 0.24 W/kg on the back. In [287], a miniaturized dual-band patch antenna was designed at 2.4 and 5.2 GHz, as shown in Figure 10b. The design was miniaturized to 74% using the combination of a DGS and a shorting via. However, dual-band was obtained using a U-slot in the ground. In [288], a metal rim antenna was proposed for dual-band applications. A T-type feeding was used to connect ground with the metal rim, whereas a shorting patch was used to excite the second antenna. The antenna was fed with a T-type feeding. Additionally, the use of a metal rim and the ground reduced the antenna up to 1500–2300 MHz with 1.6 dB gain in the entire band. The SAR on the phantom was about 0.78 and 0.75 W/kg, respectively, in [16], a compact miniaturized textile antenna was designed at 2.4 GHz, which used a rectangular slot along with an inserted strip line for the design of an inverted E-shaped antenna to minimize the size up to 75% as compared to a conventional antenna. The antenna showed a stable radiation performance, with 15% BW, and 79% radiation efficiency. In [289], a dual-band array type antenna was designed for the WiMAX, LTE, and WLAN applications. The MTM structure was used as a ground plane to reduce the size. The radiation efficiency and the gain in the first band were obtained up to 95% and 2.8 dBi, respectively, while, in the second band, the radiation efficiency and gain were achieved up to 85% and 3.48 dBi, respectively. In [290], a circularly-polarized antenna was proposed at 2.32–2.63 GHz, as shown in Figure 10d. In that context, a square type ring was asymmetrically truncated at the surrounding edges to get the circular polarization and to reduce the patch size (50.5%). Moreover, the slots and stubs were used in the ground as a DGS to further reduce the size. The antenna showed a BW of 12.53% and a gain of 5.65 dBi along with the axial ratio (AR) of up to 3.27%.

Recently, wearable designs have received special considerations due to their demands in various fields, like health and military applications. However, the textile materials have shown flexibility and conformity on-body as well as easily integrated into the clothing [291]. In [292], a soft textile surface was used to decrease the back radiation of the antenna during moving. The backward radiation was significantly reduced on a cylindrical surface. The backward radiations of 6.2% were found in a flat position without a soft surface, while that of 2.2% were obtained when the antenna was surrounded by two-unit cells. The backward radiations of 8.2% and 2.6% were obtained during a curved surface. However, the soft surface improved the radiations in the broadside direction. In [293], a novel miniature feeding structure was used in the aperture-coupled antenna at 2.4–2.5 GHz. The design reduced the size of the printed circuit board (PCB), which was carrying an electronics circuit and feeding system, as shown in Figure 11a. The feeding point was found so small that it did not affect the users. However, the main characteristics such as the cross-polarization, the FBR, and the gain were obtained up to −20 dB, 15 dB, and 5.6 dBi, respectively. Additionally, a 47% efficiency and 0.145 W/kg SAR was obtained on the body. In [294], another compact textile-based antenna was proposed for short-range communications at 5.9 GHz. The design was fully textile-based with a 9.3% BW and a gain of −2.75dBi.

In [210], a PIFA type antenna was designed from 2.40 GHz to 2.48 GHz using a reverberation chamber as a shorting wall instead of shorting pins for increased BW. Moreover, a slot was used in the patch to widen the BW. The design showed a BW of more than 46% and a gain of 1.5 dBi. In [295], a textile antenna was designed for the GSM/PCS/WLAN applications, which provided the comfortability with a gain of 2 dBi obtained in all bands. In [296], a wearable antenna was proposed for military applications at 915 MHz (ISM) and 1.575 GHz (GPS L1) bands. This design used two patches, such as truncated and circular ring patches, with four conductive threads. The circular shape was used for the ISM band, whereas the truncated patch was used for the GPS band. In [297], the authors proposed an AMC based coplanar waveguide antenna for the ISM bands, as illustrated in Figure 11b. The AMC structure improved the performance and provided the isolation between the antenna and the human body. The gains of 8.2 dBi and 9.95 dBi at 2.45 GHz and 5.8 GHz were achieved, respectively. The SAR was reduced to 96.5% for 1 g of tissues and 97.3% for 10 g of tissues. In [298], the EBG based circular ring slot antenna was proposed for the ISM band. The gain improvement of 7.3 dBi, and the efficiency of 70% was achieved. The SAR value of 13.5 W/kg for 1 g tissue and 7.98 W/kg for 10 g was obtained. Moreover, the EBG structure has reduced the SAR from 95.9% to 97.1%, and a new value of 0.554 W/KG was obtained. In [299], an M-shaped antenna was designed on a Kapton substrate at 2.4 GHz using an AMC structure as a ground plane. This antenna used an AMC structure as a ground, which was fed through a CPW. Consequently, FBR was increased up to 8 dB with a gain of 3.7 dBi as compared to the antenna without the structure. Moreover, the SAR value was reduced up to 64% and showed 0.683 W/Kg. However, a similar antenna without AMC showed up to 1.88 W/Kg. In [300], two different types of meander line antennas were designed at 915 MHz. Their shapes were like Vivaldi antennas and the edges of both antennas were connected through the meander lines. The gain, efficiency, and BW of one of them was −3.75 dBi, 31%, and 28 MHz (3%), and for the second one, it was −0.3 dBi, 76%, and 73 MHz (8%), respectively. In [301], a textile antenna was proposed at 5.8 GHz, which used a SIW cavity-feed structure with a circular ring slot. This antenna provided the gain and radiation efficiency of 2.8 dBi and 32.5%, respectively, on the voxel.

A SIW antenna has been considered as a new technology in the WBAN systems. It is highly recommended due to its low cost and the ease of design for upcoming technology, like 5G [302]. This structure used the vias on its surrounding walls and was supported by a ground. Additionally, it improves the quality factor (Q) of the design as well as provides shielding against thermal radiation between the human body and antenna. In [303], the SIW technique was implemented for designing two antennas, which were located on the same substrate. In [304], a multiband wearable antenna was designed on a leather substrate. The copper was used as a radiator, which was pasted on the leather using sticky material to operate for WiMAX, WiFi and military applications. In [305], a copper taffeta fabric was etched on a felt substrate to obtain a low profile wide-band antenna, which provided the BW and radiation efficiency of 46% and 84%, respectively. In [306], the MTM-based SIW textile antenna was designed on the same felt substrate using conductive fabric, which reduced the size up to 80% with an efficiency of 74.5% for on-body applications. The research in [307] proposed a SIW-type fully-textile antenna at 2.45 GHz. This design presented a stable performance and gave a low SAR in bending as well as integration. Therefore, the SIW type antennas are available in various shapes with suitable performance. Recently, a new miniaturized wearable mosaic antenna is designed for cross-body applications with higher performance as compared to other types of wearable antennas, as shown in Figure 11c [308]. The design optimally uses the surface waves to support cross-body communication. The mosaic antenna can be used to detect human activity with an accuracy of 91.1% using RF-type recognition methods in the WBAN systems. Thus, it can work as a wearable sensor for motion tracking and human activity recognition (HAR), simultaneously. 

Currently, micro and millimeter-wave have received great attention in communication. They can be widely used in various applications including sensing, or medical systems, etc. Thus, the terahertz (THz) technology would be able to fulfill the high demands, such as higher data rates, multiple features in one, miniaturized devices, and 5G networks within the range of 0.1–10 THz. However, the current advanced applications are not being fully used by the concerned society due to the immaturity of THz in light of antennas and some basic components [309]. Researchers in [310,311] investigated various types of THz antennas to fulfil the demands of promising applications. In [312], a graphene antenna was designed at 0.647 THz, as illustrated in Figure 11d. The antenna was tested for on-body applications. It showed a BW of 20 GHz with a radiation efficiency of 96% (50%) in free space (on-body), and a gain of 7.8 dB (7 dB), respectively.

In this section, two categories of miniaturized wearable antennas have been discussed in detail with different examples. Table 10 has been provided to show the performance comparison of these miniaturized wearable antennas. In practice, it is important to use wearable antennas with textile accessories like a button antenna because it can be easily integrated on various locations of the body. Additionally, button antennas use a combination of flexible and hard materials and cannot be easily deformed, and are less sensitive to body humidity. They provide good efficiency and gain in free space and on-body applications, including low SAR. Recently, various unique button antennas have been designed in the literature based on the principle of the chassis or terminal antenna using characteristic mode theory [275], which can easily improve the performance and reduce their size. Thus, all these advanced techniques are linked with the fast development of smart mobiles, which have reshaped the wireless communication devices [313].

## 5. Applications of Wearable Antennas

The user’s requirement for wearable devices has grown rapidly with the advent of current monitoring systems. Specifically, they are quickly increasing in the healthcare system, which will have a good impact on society. Physiological issues are essential health indicators and their monitoring could successfully allow initial detection of any kind of disease. The smart wearable applications would decrease serious health challenges, such as disease control and cost. Usually, clinical devices are non-wearable for measurement; thus, researchers have focused on the unique smart shapes that would enable wearable devices with a low profile, robust, and high-performance. The IEEE 802.15 group has recognized standard applications for body-centric communications [314]. Various types of wearable antennas can be used in the WBAN applications, such as reconfigurability, rectennas, and dielectric resonating antennas [315,316,317,318]. The demand for smaller wearable devices is increasing, especially in daily-use appliances. The smaller wearable devices require unique smart designs, which means that the antennas will work with the small ground using different shapes and techniques. These techniques present a challenge to researchers. Moreover, the wearable devices span the gamut from unique smart rings, button, bracelets, and necklaces, to smart glasses and watches, to smart gloves, socks, and t-shirts, or different decorative clothing accessories or other ornaments, etc. A brief discussion about the unique smart designs of the wearable antennas is provided in this section [185].

Unique Smart Designs of Wearable Antennas

The wearable antennas are required to be sewed on clothes, garments, or to be integrated into personal accessories, such as shoelaces, shoes, glasses, buttons, shirts, coats, and helmets. Subsequently, these techniques have various advantages, such as robustness, limited space, and low cost. Additionally, such designs are also suitable for rigid material like FR4, roger, etc. Moreover, button, wristwatch, shoe, or hearing instrument usually used copper or silver, as the conductivity of such materials is more than the smart textile or other flexible conductive materials. In [319], a 3D printing and inkjet printing methods were used for RFID applications. Furthermore, the RFID tag provided the wireless connectivity within the clothes for the authentication of a wearer. In [320], a flexible antenna was proposed to integrate with the Louis Vuitton logo at 2.45 GHz and 4.5 GHz. The logo shape was based on four arms, two longer-type arms for lower frequency, and two shorter and thick-type arms for higher frequency. The impedance BWs and gains of 17.1% and −0.29 dBi obtained for a lower band, and 13% and 3.05 dBi obtained for the higher band, respectively. A unique smart logo-shaped antenna has been designed using a non-woven conductive fabric on a jean’s substrate at 1.8 GHz. The design showed a bandwidth of 70% along with a dipole type radiation pattern [321]. Various types of colorful textile antennas have been designed that are highly flexible, conformal, lightweight, robust, and show similar performance copper counterparts. In [322], various embroidery tag antennas, like Korean letters UHF RFID tag antennas, have been fabricated at 910 MHz, as depicted in Figure 12a. The S_11_ is nearly −13 dB along with 17 MHz B.W. In [223], Levi’s logo-type antenna was designed for remote monitoring at 2.53 band. The logo shape was designed using a non-woven smart textile with a cutting plotting method. In [323], a new wearable metallic structure was proposed for WBAN applications, as shown in Figure 12b. In this design, a metallic belt buckle was used to obtain an antenna embedded in a belt at the ISM and WLAN bands. The gain of 2.8 dBi was obtained at 2.45 GHz and 4.5 dBi at 5.25 GHz. Additionally, due to its unique shape, it could not be affected by any bending conditions. In [324], a monopole antenna in a hang bag zip was proposed, as shown in Figure 12c. The results showed consistent performance on hand phantom with a gain of 5 dBi at 2.4 GHz. Moreover, a zipper shape was found to be easily sewed within flexible materials like jean or denim materials. In [325], a shoelace antenna was investigated using a thin, flexible wire for the collision avoidance of a blind person, as shown in Figure 12d. The proposed design was fixed at the downside of laces and it showed an acceptable performance during the loosing or tightening of laces.

In [326,327,328], researchers investigated a strip watch antenna, which acted as the main radiator. A wristwatch was used as a conductive ground like a cavity-backed annular slot antenna at 2.4 GHz. Additionally, the conductive ground minimized the backward radiations and improved radiation efficiency up to 66% on a hand model. In [329], a MIMO antenna was designed on a plastic wristband strap at 5.2–5.8 GHz. The antenna achieved an efficiency of 97% in free space and 55% on the hand model. In [330], a PIFA antenna was designed in a Bluetooth transceiver, which was integrated with a wristwatch casing. In [331], a worn finger antenna was investigated at 25–35 GHz to obtain a circular polarization on a wearer’s fingers, which showed an end-fire radiation for on-body antenna, with a good CP obtained during finger tilting. Similarly, several examples of wearable antennas on rings are proposed in [332,333]. In [334], a transparent antenna was placed on glasses for medical purposes. It works in a way that when a patient wears the glasses, the eyes can feel and measure pressure. This antenna design used multi-layered substrates, such as indium, zinc, and tin oxide (IAI) film (σ: 2 × 10^6^ S/m), with an efficiency of up to 46.3% obtained on the phantom. In [335], a dual-band antenna was designed on the plastic eye frame at 2.45 GHz. An efficiency of 22.8% was achieved on the head model. In [149], an array-type of wearable antenna was investigated on an ultra-thin Taconic substrate at 60 GHz. The design showed a directional radiation with a gain of 14.8 dBi along the boresight. Researchers in [336] designed a slotted antenna in a glove for UHF RFID, with a realized gain of −1.3 dBi at 866 MHz. In [215], a novel spiral antenna was designed on conductive threads at 0.3–3 GHz. Moreover, the design was tested in free space and under a bending condition, and it provided nearly CP with a gain of 6.5 dBi. In [337], a novel monopole antenna was designed on fiber at 2.4 GHz. This antenna acted as multiple internal electrodes and sensed in all three dimensions. Overall, its performance was found better than the conventional copper-base antenna. In [338], an electrically small antenna was designed for a hearing instrument (HI) at 2.4 GHz, as shown in Figure 12e. This design was based on a cavity-backed shape using a 3D-printed substrate, and it occupied 40% volume (a: 12 mm) of an ear. However, the gain and the efficiency of 1.54 dB and 70% were obtained at 2.51 GHz, respectively, whereas the gain and efficiency of 6.50 dBi and 22% were obtained at 2.45 GHz with a limited SAR value.

In [339], a new shape of a dipole array antenna was proposed for integration with clothes and shoes to monitor the safety and tracking activities at 2.45 GHz. The antenna was fabricated on a laminate substrate (ε_r_: 3.1, tan δ: 0.0026) with a copper-foil radiator (18 µm). The antennas obtained a 15% impedance BW at 2.45 GHz. Moreover, the design was miniaturized so that its performance could not be affected during bending and it provided limited SAR. Similar, for health care, wearable electronics can realize the functions of various kinds of communication and entertainment devices, like an mp3 player made in the shape of a bangle or Bluetooth headset, are conceptualized [340]. In [341], the antenna was designed for a sports shoe at 6–8.5 GHz. The design was tested on the foot phantom. Additionally, the simulation and measurement results showed that the antenna could give stable performances in the required band. The human hand can be used as an antenna, called hantenna, as shown in Figure 12f. This technique was proposed in [342], where a wireless link between an antenna and the human hand was used. Experimentally, it was shown that the human body, especially the hand, even being a high lossy medium, could be useful for the antenna design. However, in this work, the signal strength for the hantenna was not the primary concern, because plenty of power is available near a mobile station. Moreover, the hands and the fingers enhanced the link-level up to 15–20 dB without any antenna (small mismatch connector) at 1.15 GHz.

## 6. Conclusions and Future Direction

A wearable antenna is a key component and required to be portable, low profile, low power, flexible, and lightweight in WBAN systems. This paper discussed up-to-date wearable antennas in light of materials and fabrication methods. Every section is thoroughly discussed, and all of the features regarding the materials and fabrication methods are compared in different tables. Additionally, two types of miniaturized wearable antennas, such as button-shaped and single and multi-band antennas, are briefly discussed with examples. Finally, unique applications of the wearable antennas, such as smart shapes, are discussed in detail with references to know the real meaning of wearable technology. Thus, this review paper reflects the current state-of-the-art.

In order to cope with the above issues, recently high-performance conductive graphene and MXene ink have been used for the wearable antennas due to their remarkable electrical properties. However, their performance is comparatively low due to their inherent skin depth issue. Several techniques have been considered, like embedding conductive materials to realize a high-performance antenna, but their performance is relatively low. Recently, a transition metallic niobium diselenide (NbSe_2_) is used for the antenna design that provides a sub-micrometer thickness of 855 nm, which is less than the skin depth of any other materials. NbSe_2_ can be the right candidate for a high-performance flexible and miniaturized wearable antenna using the spray-painting method. These transition metals have opened new research areas in wearable antennas, such that further research is required for the extraction of human-tissue parameters and body models to achieve a high-performance antenna. The improvement in the fabrication method is also necessary to maintain the antenna performance in more adverse conditions, like the oxidation (redox) process. Thus, this paper covers all the salient aspects of wearable antennas in the light of materials, fabrication methods, designs, and their unique smart shapes applications. This comprehensive review paper has managed to move the wearable antenna designs into a particular direction. This paper will be beneficial for researchers to produce high-performance, low-cost, and mass-produced wearable antennas for further applications.

## Figures and Tables

**Figure 1 micromachines-11-00888-f001:**
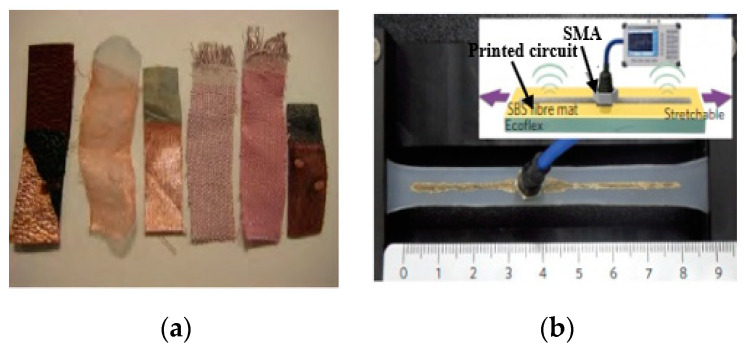
(**a**) Different conductive textile material [96], (**b**) conductive composite materials [135].

**Figure 2 micromachines-11-00888-f002:**
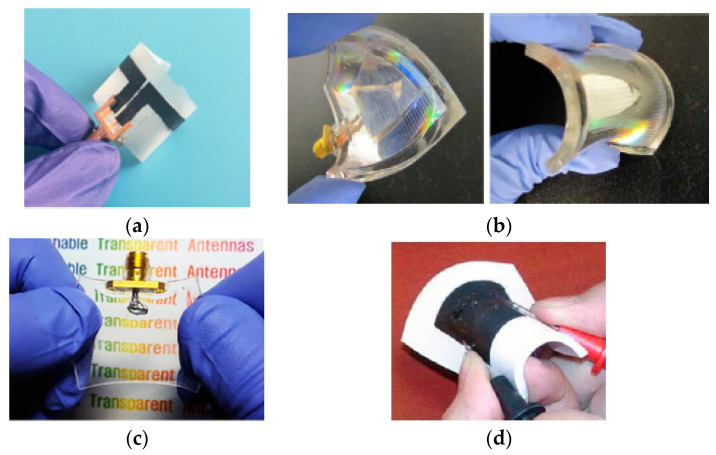
(**a**) Lycra fabric on porous film [153], (**b**) flexible antenna composed of liquid metal encased in the elastomer [154], (**c**) Ag/NW embedded PDMS [155], and (**d**) fabric-embedded PDMS [144].

**Figure 3 micromachines-11-00888-f003:**
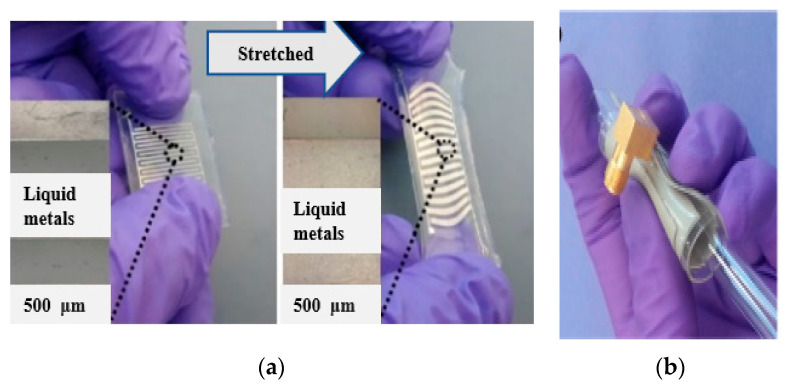
Different flexible materials. (**a**) Stretchable PDMS [171], (**b**) rolled patch antenna [173].

**Figure 4 micromachines-11-00888-f004:**
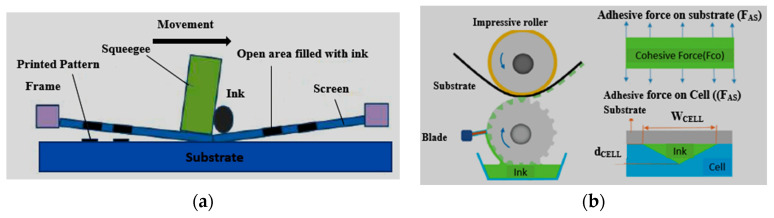
(**a**) Fabrication process for multilayer fabrication [177], and (**b**) gravure printing [185].

**Figure 5 micromachines-11-00888-f005:**
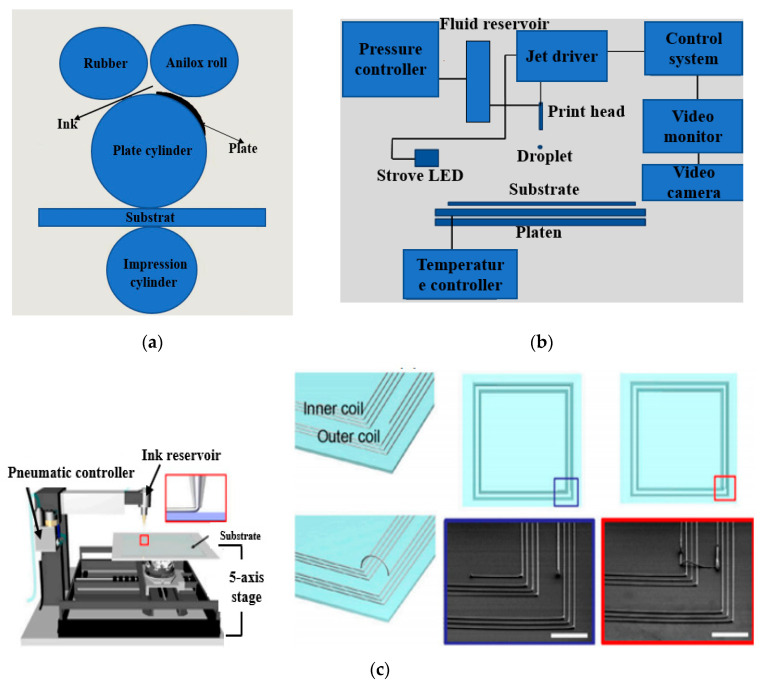
(**a**) Flexography printing process [207], (**b**) inkjet printing process [177], and (**c**) 3D direct ink writing method [116].

**Figure 6 micromachines-11-00888-f006:**
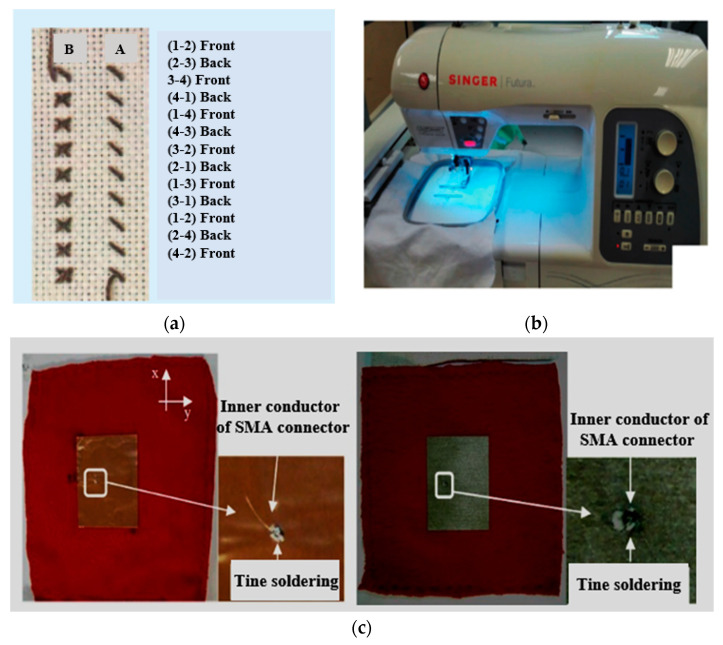
(**a**) Fabric used to realize the patch employing conducting threads and subscripts “back” and “front” of the fabric [211], (**b**) embroidery machine [213], and (**c**) adhesive copper tape (adhesive conductive fabric) [211].

**Figure 7 micromachines-11-00888-f007:**
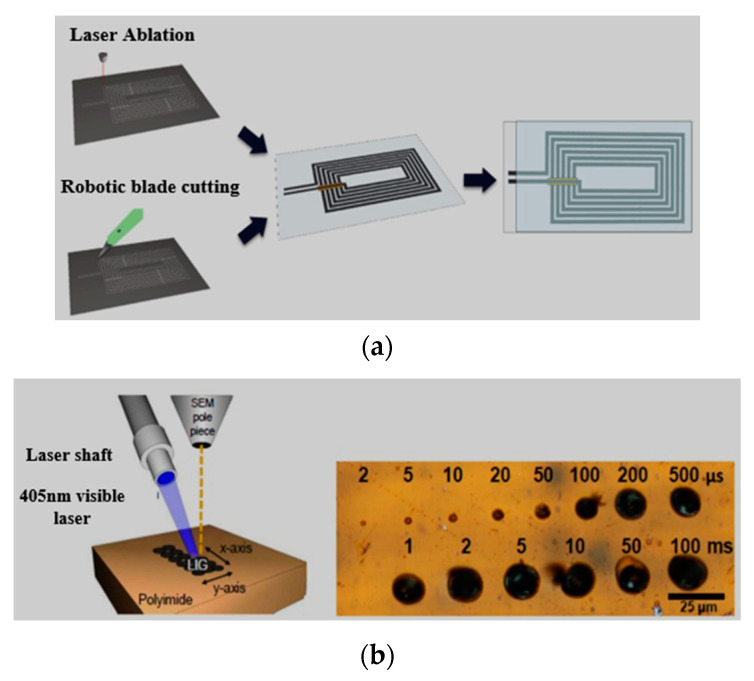
(**a**) Graphene antenna [174], and (**b**) infrared laser method (LIG) [227].

**Figure 8 micromachines-11-00888-f008:**
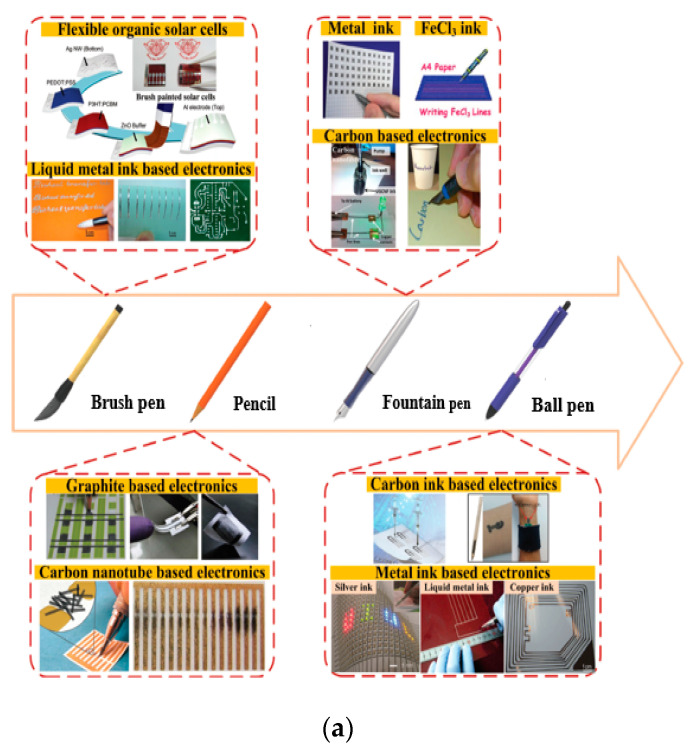
(**a**) Development in direct-writing electronics [260], (**b**) design of a spray painting method (Drexel’s spray paint antenna) [116,256].

**Figure 9 micromachines-11-00888-f009:**
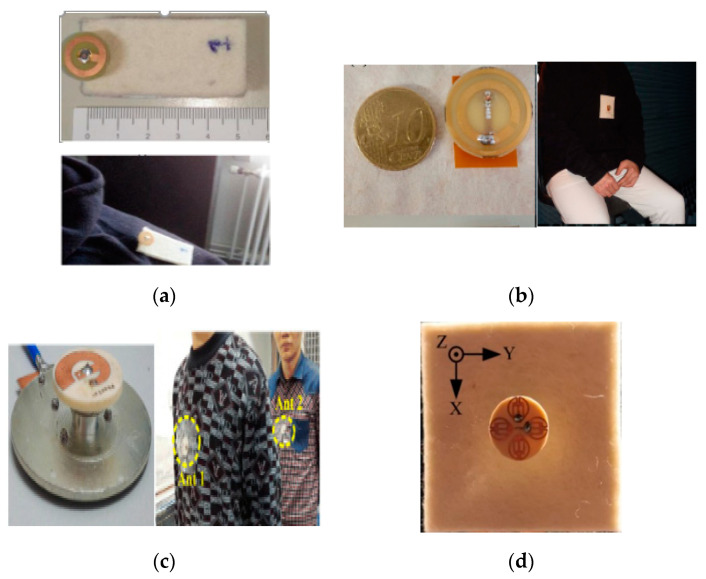
(**a**) Wideband button [275], (**b**) circularly-polarized button [276], (**c**) F-inverted button [279], and (**d**) array button antenna [280].

**Figure 10 micromachines-11-00888-f010:**
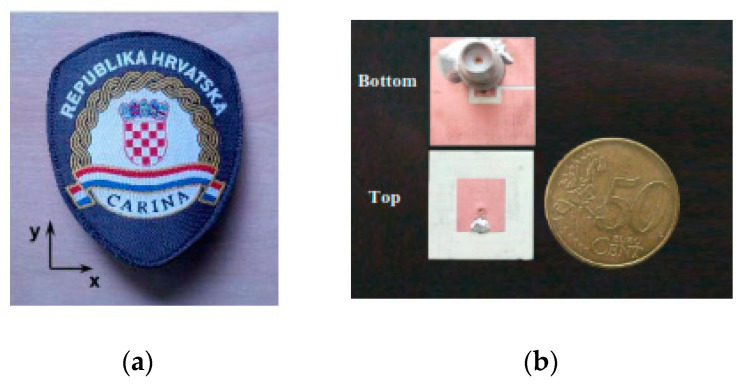
(**a**) Sleeve-badge antenna [285], (**b**) miniaturized dual band antenna with DGS [287], (**c**) E-shaped patch antenna [16], and (**d**) circular-polarized antenna [290].

**Figure 11 micromachines-11-00888-f011:**
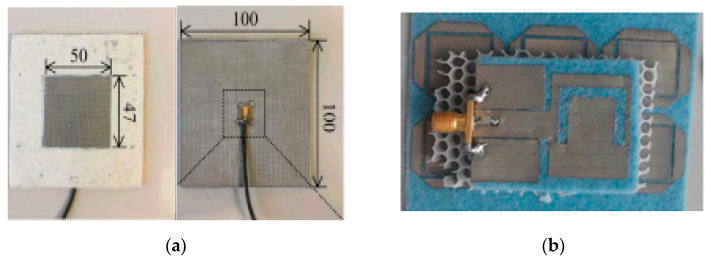
(**a**) Patch antenna with miniaturized feeding network [293], (**b**) AMC antenna [297], (**c**) mosaic antenna [308], and (**d**) THz graphene antenna [312].

**Figure 12 micromachines-11-00888-f012:**
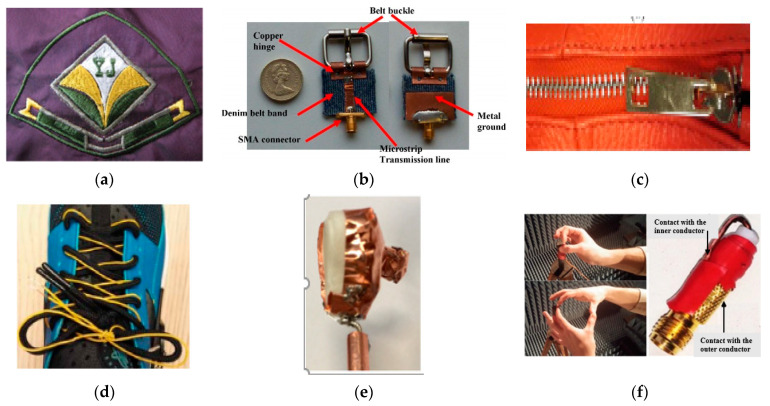
(**a**) Korean letters logo shape [322], (**b**) metallic belt antenna [323], (**c**) zip antenna [324], (**d**) shoelace antenna [325], (**e**) shell antenna [338], (**f**) human hand antenna (hantenna) [342].

**Table 1 micromachines-11-00888-t001:** Applications of wearable antennas [63].

Fields	Applications
Healthcare	Bio-signals such as ECG, EEG, blood pressure, glucose level, breast cancer detection, doppler radar, smart home, telemedicine.
Entertainment	Sports, gaming, ear instrument, smartwatches, jackets, smart shoes.
Military	Battlefield, shoes, coat, helmet, identification, etc.

**Table 2 micromachines-11-00888-t002:** Flexible conductors [96].

Conductive Materials	Thickness (t: mm)	Conductivity (σ: s/m)
Egaln Liquid	0.08	2.5 × 10^5^
Polyleurethene Nanoparticle Composite	0.0065	1.1 × 10^6^
Zoflex + Copper	0.175	1.93 × 10^5^
Silver Flakes Fluorine Rubber	NA	8.5 × 10^4^
AgNW/PDMS	0.5	8.1 × 10^5^
PANI/CCo Composite	0.075	7.3 × 10^3^
Copper Coated Taffeta	0.15	3.4 × 10^6^
Graphene	100 μm	33 × 10^3^

**Table 3 micromachines-11-00888-t003:** Properties of the substrate [96,176].

Materials	Dielectric Constant (ε_r_)	Loss Tangent (tan δ)
PTFE	2.05	0.0017
PDMS	3.2	0.01
Ethyl Vinyl Acetate (EVA)	2.8	0.002
PDMS Ceramic composite	6.25	0.02
Neoprene rubber	5.2	0.025
Kapton Polyimide	3.4	0.002

**Table 4 micromachines-11-00888-t004:** Comparison of different flexible materials [177].

Properties	Polymer	Textile	Paper	Fluidic
Dielectric Loss/Loss Tangent	Low Loss	Low Loss	Medium Loss	High Loss
Tensile Strength	High (165 MPA)	Low (2.7 MPA)	Low (30 MPA)	Low (3.9 MPA)
Flexural Strength	High (p.s.i)	Low (8900 p.s.i)	Low (7200 p.s.i)	Low (650 p.s.i)
Deformability	Low	High	High	High
Thermal Stability	High	Low	Low	Low
Fabrication Complexity	Simple/Printable	Complex	Simple/Printable	Complex/Non-printable
Robustness to Wetness	High	Medium	Low	Low
Cost of Fabrication	Medium	Low	Low	High
Weight	Medium	Low	Low	Medium
Overall Size	Small	Large	Large	Small
Stability for Integrated Circuit	High	Low	Low	High

**Table 5 micromachines-11-00888-t005:** Comparison of different printing methods [185].

Features	Screen	Gravure	Flexographic
Printing Forms	Stencil/R2R	R2R	R2R
Approach of Printing	Contact	Contact	Contact
Printing Speed (m/min)	10–15	100–1000	100–500
Line Width (lm)	30–50	10–50	45–100
Surface Tension (mN/m)	38–47	41–44	13.9–23
Resolution (µm)	30–100	50–200	30–80
Speed (m/min)	0.6–100	8–100	5–180
Substrate	Flexible	Flexible	Flexible
Process Mode (step)	Multiple	Multiple	Multiple
Mask Requirement	Yes	No	No
Material Wastage	Yes	Yes	Yes
Ink Composition (wt%)	-	-	-
Pigment	12–20	12–17	12–17
Binder	45–65	20–35	40–45
Solvent	20–30	60–65	25–45
Additive	1–5	1–2	1–5

**Table 6 micromachines-11-00888-t006:** Comparison of different inkjet or direct ink writing methods [208].

Methods	Ink Viscosity (cp)	Nozzle Diameter (μm)	Size (μm)	Thickness (μm)	Features
IJP	1–30	20–60	20–100	0.6	Low-cost, multiple heads
AJP	1–1000	150–300	10–200	0.1	High throughput, thin layers, good features
3D	1–1,000,000	0.1–1	1–1000	0.5	Most precise, best resolution

**Table 7 micromachines-11-00888-t007:** Comparison of fabric and adhesive techniques [221].

Features	Embroidery	Weaving	Adhesive
Commercially Available	Yes	Yes	Yes
Extra Processes (cutting)	Yes	Yes	Yes
Waste Product	Yes	Yes	Yes
Fabrication Accuracy	High	Medium	Medium
Material Cost	Medium	Low	Low
Easy to Attach	Yes	Yes	Yes

**Table 8 micromachines-11-00888-t008:** Comparison of direct cutting [174,228,229].

Features	Laster Cutting (Trotec Speedy 300)	Robotic Cutting Plotter (Silhouette Cameo)
Commercially available	No	Customized by Requirement
Extra Processes	No	No
Waste Product	No	Yes
Fabrication Accuracy	High	Medium
Material Cost	High	Low
Optimum applications	Small, Complex Patterns Required High Resolution	Customized and Large Size

**Table 9 micromachines-11-00888-t009:** Comparison of handwriting methods [260].

Writing Methods	Writing Materials	Substrates	Conductivity [S cm^−1^]	Width [μm]
Brush Pen	PEDOT:PSS/P3HT/PCBM, Silver Nanowire Ink, CNT Ink, V2O5 Ink, Gallium Ink	Glass, PET, ITO, Paper	50–2.9 × 10^4^	50–500
Pencil	Graphite Rod, Pencil Lead, SWCNT/MWCNT	Paper, Salt	20–884	900–1900
Ball-Pen	Different Inks such as Silver, Copper, Gallium Enzymatic	Paper, PDMS, Skin	5 × 10^3^–1 × 10^5^	100–800
Fountain Pen	Carbon Nanofibre Ink, CNT Ink, FeCl_3_ Ink	Paper	-	770–980
Spray	Different Inks/Liquids such as Silver, Copper, Gallium and Enzymatic	Any	5 × 10^3^–1 × 10^5^	100–800

**Table 10 micromachines-11-00888-t010:** Comparison of different wearable antennas.

**Examples: Comparison of Button Antennas**							
**Ref**	**Button Antenna (Patch-Size)** **(mm)**	**Frequency (MHz/GHz)**	**Antenna Structure**	**Substrate Materials** **(ε_r_/tan δ)**	**Conductor** **(σ or** **Rs)**	**SAR (W/kg)**	**Radiation Efficiency (total Efficiency) % or Gain (dBi) or Other Performance**
[278]	40 × 40	2.4/5.3/8	Modular Snap-on Button	Cuming-Foam PF4	Shieldit	NA	7.8/3.1 (Half-Wave) 8.9 (Quarter-Wave)
[275]	Dia(d): 15	2.4	Circular Button with Split-Ring Resonator (CSRR)	Button Disc, FR4 Ground Substrate, Felt	Copper/Shieldit (Patch/Ground)	0.45	Free Space, On-body 97, 71 or 1.8/5.1
[276]	Dia (d): 9.77	5.5	Circular Button	Button Disc, FR4, Ground Substrate, Felt	Copper/Shieldit (Patch/Ground)	0.123	Free Space, On-body 79.9, 70.8 or 3.5 (Entire band)
[272]	Dia (d): 16	2.4/5	Circular Button	Button Disc Rogers 5880 Ground Substrate Felt	Copper/Shieldit (Patch/Ground)	On chest 0.18/0.12 On arm 0.20/0.22	Free Space, On-Body at Lower/Upper band ≥ 90/84 or 1.05, 0.24/4.50, 4.73
[277]	Dia (d): 22.3	2.4/5.6	Circular Cuff Button	PTFE Ceramic	Copper (Patch/Ground)	NA	Free Space (94) or 1.5 (Both Bands)
[280]	Dia (d): 19.5	4.50–4.61/ 5.04–5.50	Circular Array Button	Button Disc RO4003	Copper/Shieldit (Patch/Ground)	On chest 0.0664/ 0.0942	Port-1 (Total-Efficiency) (86/93) or (7.4/7.6) Port-2 (Total-Efficiency) 85 /92 or (7.5/7.4)
[279]	Radius (r): 18	2.45/5.8	F-Inverted Button	Button Disc Rogers4003	Patch/ Flange Ground (Copper)	1g, 10 g 0.370, 0.199/0.584, 0.232 (Sim)	On-body 46.3/69.3 or −0.6 and 4.3 dBi
**Examples: Comparison of Single and Multiband antennas**							
**Ref**	**Antenna (mm)**	**Frequency Range (MHz/GHz)**	**Antenna Structure**	**Substrate Materials** **(**ε**_r_/tan δ)**	**Conductor** **(σ or Rs)**	**SAR (W/kg)**	**Radiation Efficiency (%) or Gain (dBi) or** **other performance**
[299]	30 × 45	2.45	M-shaped with AMC	Kapton Polyimide	Silver Ink	0.683	3.7
[343]	59.8 × 59.8	2.3 to 2.68	Reconfigurable Patch	PDMS	Shieldit	For 10 g 0.282	2.6
[344]	70 (dia)	2400/5800	Circular Patch	PDMS	Shieldit	0.248/0.091	On-body Lower, Upper band 58.6, 50.6 or 4.16, 4.34
[306]	74.5 × 48 × 3.34	2.45	SIW Slot	Wool Felt	ShieldIt Super	0.380 (Max)	On-body 74.3 (Total Efficiency) or 5.35
[345]	80 ×61 ×4.51	3.1–10.6	UWB	Felt	ShieldIt	1.21 (3GHz), 0.52 (7GH)	On-body 96.6, 99.6
[285]	50 ×57	2.45	Circular Patch	Multilayered Textile	Shieldit	0.34	55.3% (Sim)
[292]	13 cm × 13 cm	4	Circular Patch with Soft Surface	Felt	Shieldex Zell RS	6.2 to 2.2% (Reduced Back radiation)	NA
[346]	52 × 250	450 MHz	Koch Fractal Dipole	Jeans	Copper Foil	0.119	2 dBi
[143]	0.5 × 25.4	1.91	Dipole	PDMS (2.67/0.0375)	EGaIn	NA	90%
[289]	14.8 × 37.8	2.11–3.05/5.2	Array Shape with Metamaterials	RT/Duroid5880	Copper	NA	First band 95 or 2.8–3.48 Second band 80
[324]	20 × 20	2.44	Handbag Zipper	FR4	Copper	NA	97/5(2.90 GHz)/97.4 (2.70 GHz)
[327]	10 × 10	2.46 to 2.5	Watch Strap	ArlonDi880	Copper	NA	5 dBi
[347]	50.0 × 37.5	2.45	Dipole	Paper	Silver Ink	NA	NA
[348]	Π × 242 × 3.2	2.5/5.8	Circular (Reconfigurable)	FR4	Copper	5.04 and 14.79 (FS)	−3.03/3.83
[349]	50 × 50	2.4	Circularly with AMC Structure	PDMS	AgNWs Composite	0.13 to 0.18	79 or 5.2
[320]	22.3 × 22.5	2.45/4.5	Louis Vuitton Logo	Leather	Shieldit	NA	Lower, Upper bands 15.2, 14.3 or −0.29, 3.05
[288]	40 × 40		Metal Rim	FR4	Copper	0.78	On-Body 50 or 1.6
[350]	4.98 × 48.96	3.3, 3.4	E-shaped	Felt	ShieldIt	NA	2.98, 4.56
[52]	3 cm × 3 cm	2.86–2.92	Stretchable	Soft Silicone	Copper	NA	NA
[351]	300 × 230	2.4	Chinese Symbols	RO4350B	Copper	NA	8.2
[257]	10 mm (L)	860 MHz	Dipole	PET	MXene Ink (5000 to 10,000 S/cm), thick, 62 nm to 8 µm)	NA	S_11_, −65 dB
[258]	10 × 10	2.01 to 2.80	Monopole patch	PET	Niobium Diselenide (855 nm thickness)	NA	70.6 or S_11_, −46.5 dB
[352]	15 × 63	2.45	Dipole	Elastomer	Graphene	NA	NA
[154]	40 × 40	3.45	Patch Antenna	PDMS	EGaIn	NA	60
[353]	53 × 63 60 × 50	868 MHz	Tattoo Antennas	Thin Adhesive Polymer	Conductive Ink	NA	60 or S_11,_ −17.1, −19.9 dB
[354]	42.4 × 48.5	2.45	Patch Screen Printed (Washable)	Cotton/Polyester (CO/PES)	Flectron	NA	S_11_, −25 to −30 (2.4) (No-washing) S_11_, −30, −35(2.5) (Few Cycle Wash)
[113]	57 × 13	UHF	RFID Tag (washable)	Cotton/Polyester (CO/PES)	Multifilament	NA	Stretchable Tag
[355]	30 × 22 µm	2.67 to 2.92 THz	Rectangular Patch	Silicon Nitride (Si_3_N_4_)	Graphene	NA	87 or 5

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
