# Peer review of "Recent Advances of Wearable Antennas in Materials, Fabrication Methods, Designs, and Their Applications: State-of-the-Art"

_micromachines, 2020, doi:10.3390/mi11100888_

Round 1

Reviewer 1 Report

A comprehensive review of art-of-the-state design and fabrication methods of wearable antennas is presented. The paper is well organized.  

Author Response

Please find attached the response to reviewer 1 report. Actually, I can not upload the revised manuscript now, because in a system, the option is closed. Once the option to upload will be opened. I will upload the revised manuscript.

Reviewer 2 Report

The authors have prepared a nice review paper for wearable antennas, etc. The only suggestion this review has is if the authors can include a table which compares relevant antennas and their performances; at this stage the comparison is geared towards different technologies which is useful. More can be done to complete the paper.

Author Response

Please find attached the response to reviewer 2 comments report. Actually, I can not upload the revised manuscript now, because in a system, the option is closed. Once the option to upload will be opened. I will upload the revised manuscript.
